# Acetylation Modification, Characterization, and Anticomplementary Activity of Polysaccharides from *Rhododendron dauricum* Leaves

**DOI:** 10.3390/polym14153130

**Published:** 2022-07-31

**Authors:** Zhengyu Hu, Jinfeng Sun, Long Jin, Tieqiang Zong, Yuanqi Duan, Hongli Zhou, Wei Zhou, Gao Li

**Affiliations:** 1Key Laboratory of Natural Medicines of the Changbai Mountain, Ministry of Education, Yanbian University, Yanji 133000, China; huzhengyu1994@163.com (Z.H.); jfsun@ybu.edu.cn (J.S.); longjin0209@163.com (L.J.); ztq405955580@163.com (T.Z.); duanyuanqi163@163.com (Y.D.); 2Engineering Research Center for Agricultural Resources and Comprehensive Utilization of Jilin Provence, Jilin Institute of Chemical Technology, Jilin 132022, China

**Keywords:** polysaccharides, *Rhododendron dauricum* leaves, acetylation modification, anti-complementary activity

## Abstract

This study focuses on the acetylation modification of polysaccharides from *Rhododendron dauricum* leaves (RDPs) with a high degree of substitution (DS) and then discusses their characterization and biological activity. The optimum acetylation conditions of RDPs were optimized by response surface methodology, which were reaction time 3 h, reaction temperature 50 °C, and the liquid-solid ratio 16 mL/g. Under the optima schemes, two eco-friendly acetylated polysaccharides from *R. dauricum* leaves (AcRDP-1 with DS of 0.439 ± 0.025 and AcRDP-2 with DS of 0.445 ± 0.022) were prepared. The results of structural characterization showed that the AcRDP-1 (9.3525 × 10^3^ kDa) and AcRDP-2 (4.7016 × 10^3^ kDa) were composed of mannose, glucose, galactose, and arabinose with molar ratios of 1.00:5.01:1.17:0.15 and 1.00:4.47:2.39:0.88, respectively. Compared with unmodified polysaccharides, the arabinose content and molecular weight of the two acetylated polysaccharides decreased, and their triple helix conformation disappeared, and further improved their anticomplementary activity. The two acetylated polysaccharides showed stronger a complement inhibition effect than the positive drug by blocking C2, C3, C4, C5, C9, and factor B targets in the classical and alternative pathways. This research indicated that acetylation modification could effectively enhance the anticomplementary activity of RDPs, which is beneficial for the development and utilization of *R. dauricum* leaves.

## 1. Introduction

As a widely cultivated medicinal plant, *Rhododendron dauricum* L. belongs to Ericaceae family and is mainly distributed in Northeast China, North Korea, and Russia [1]. The leaves of *R**. dauricum* are of high economic value as a medical herb, containing various chemical constituents (e.g., volatile oil, polysaccharides, flavonoids), and is commonly applied for the treatment of pulmonary infectious diseases (e.g., cough, asthma, acute, and chronic bronchitis) [2]. Polysaccharides as the main bioactive component of *R. dauricum leaves*, our previous study showed that the two neutral polysaccharides that were derived from *R. dauricum* leaves (RDP-1: 1.0948 × 10^4^ kDa and RDP-2: 1.1899 × 10^5^ kDa) were composed of mannose, glucose, galactose, and arabinose [3]. However, the antioxidant activity of RDP-1 and RDP-2 is weak, which is unsatisfactory for the utilization of polysaccharides from *R. dauricum* leaves (RDPs). Therefore, the investigation on the modification of polysaccharides and their new biological activity may contribute to the development of RDPs.

Recently, the modification to natural polysaccharides has been proposed as an approach to enhancing the biological activities or inducing entirely new activities through a change to the substituents, which has attracted increasing attention [4]. Acetylation modification has become one of the common methods of polysaccharides chemical structure modification due to its advantages of fast reaction speed, mild reaction, and high yield [5]. Previous studies by Xie et al. and Hitri et al. showed that acetylated polysaccharides have strong biological activities, because acetyl groups can improve the molecular chain extension, molecular weight, and conformational properties of polysaccharides, resulting in the exposure of polysaccharide hydroxyl groups and affecting their biological activity [6,7,8]. Thus, acetyl modification is considered an ideal solution to improving the biological activity of RDPs. In spite of this, there is still no report that has been published on the modification process, characterization, and activity of acetylated RDPs.

The complement system refers to a set of more than 30 proteins that function as a significant effector in the immune system and contribute to the first-line host defense [9,10]. However, excessive activation and dysregulation of the complement system can lead to a variety of autoimmune diseases [11,12,13]. In particular, several reports have shown that patients with severe COVID-19 show prominent complement activation in their lungs [14,15], which suggests the potential of the complement being closely related to pulmonary infectious diseases. Simultaneously, the medicinal plant polysaccharides have also been demonstrated to be important raw materials for novel complement inhibitors with low toxicity [16,17,18]. By taking into account the traditional efficacy of *R. dauricum* leaves in the treatment of pulmonary infectious diseases and the potential activity value of its polysaccharides, it is considered significant to investigate the anticomplementary activity of RDPs and acetylated RDPs.

Therefore, this study is aimed to obtain the acetylated RDPs with a high degree of acetyl substitution by optimizing the acetylation modification process, which is beneficial to improve the anti-complementary activity of RDPs. The acetylation modification process of RDPs was optimized by response surface methodology. The two acetylated polysaccharides (AcRDP-1 and AcRDP-2) were obtained under the optimum conditions, and then characterized by Fourier infrared spectroscopy, high performance liquid chromatography, high performance gel chromatography, and UV-vis spectrophotometry. Besides, the anticomplementary activity of the acetylated polysaccharides were studied and compared with unmodified polysaccharides, to further identify their specific targets in the complement system, which provided a theoretical basis for the application of *R. dauricum* leaves in the pharmaceutical industry.

## 2. Materials and Methods

### 2.1. Materials

The dried leaves of *R. dauricum* L. were obtained from Yanji city, Jilin Province, China, and authenticated by Professor Gao Li, and the voucher specimen (YB-RD-201610) has been deposited at the College of Pharmacy, Yanbian University.

All chemicals and solvents that were used in study were analytical grade. Heparin was purchased from Coolaber Technology Co., Ltd. (Beijing, China). Sheep and rabbit red blood cells were purchased from Sbjbio (Nanjing, China). Complement reagents were purchased from Complement Technology, Inc. (Shanghai, China). Normal human serum was collected from healthy adult volunteers. On 18 September 2020, the Yanbian University Institutional Research Ethics Committee accepted all the experimental procedures and protocols (approval NO. YBU-2020-091801).

### 2.2. Preparation of Polysaccharides and Their Acetylated Derivatives

The polysaccharides from *R. dauricum* leaves (RDPs) were extracted and isolated according to the description of our previous work [3]. Briefly, the dried *R. dauricum* leaves were extracted with distilled water (30:1 mL/g) at 85 °C for 1.5 h. Then, the crude RDPs were distilled with water, and decolorized with D101 macroporous adsorption resin, and deproteinized with the Sevag solvent. Subsequently, the purified polysaccharides were re-dissolved, and subjected to the DEAE-52 cellulose column which was eluted with deionized water and 0.1 mol/L NaCl at a flow rate of 1.0 mL/min. The two fractions were dialyzed, freeze-dried, and named as RDP-1 and RDP-2, respectively.

The acetylated polysaccharides were prepared according to previous literature with some modifications [19]. In detail, the polysaccharides were dissolved in distilled water, and the pH of the polysaccharide solution was adjusted to 9 with 0.5 mol/L NaOH solution. Then, acetic anhydride was gradually added into the polysaccharide solution, and NaOH solution was added simultaneously to maintain the pH value of the reaction solution at 8~10. The reaction was carried out at a certain time with water bath temperature, and terminated by adding 1 mol/L HCl until pH = 7. Finally, the acetylated polysaccharide solutions were dialyzed in a dialysis bag (molecular weight cut-off: 3500 Da), concentrated, and lyophilized to obtain acetylated polysaccharides for further research.

### 2.3. Degree of Substitution Measurement

The acetyl group was determined by the hydroxylamine hydrochloride method [20]. The different concentrations of *β*-D-glucose pentaacetate standard solution were mixed with hydroxylamine hydrochloride (0.1 mol/L), NaOH (1.5 mol/L) and HCl solution (2.0 mol/L) in turn, and placed at room temperature for 20 min. Then, FeCl_3_ solution (0.4 mol/L) was added and reacted for 10 min. The absorbance was recorded by UV-visible spectrophotometer at 500 nm. The standard curve of the acetyl content was prepared with acetyl mass concentration as abscissa and absorbance as ordinate. Similarly, the absorbance of the sample was measured at 500 nm and substituted into the standard curve equation to calculate the acetyl content of the sample (W). The degree of substitution (DS) of polysaccharides was calculated according to the method described [21], as follows:(1)W (%)=(C×V×N)/M×100%
(2)DS=(162×W)/[4300−(43−1)×W]
where C is the concentration of the acetyl group in the sample solution (mg/mL), V is the volume of sample solution (mL), N is the dilution multiple, and W is the mass of the sample (mg).

### 2.4. Single Factor and Response Surface Design of Acetylation Modification

There were three factors that were evaluated to research the effects on the degree of substitution (DS) of polysaccharides: the ratio of acetic anhydride and polysaccharides was 8, 12, 16, 20, and 24 mL/g; the reaction time was 1, 3, 5, 7, and 9 h; the reaction temperature was 30, 40, 50, 60, and 70 °C; and the DS was used as the evaluation index. Based on the single factor experimental, the reaction time, reaction temperature, and the ratio of acetic anhydride solution to polysaccharide powder (i.e., liquid-solid ratio) were determined as three independent variables (marked as variables X_1_, X_2_, and X_3_) with three levels for each variable (coded levels as −1, 0, and 1). The Box–Behnken Design (BBD), variance analysis (ANOVA) and regression model were carried out using design expert software version 8.0.6.1 (Stat Ease Inc., Minneapolis, MND, USA). A quadratic polynomial Equation (3) was used to predict the optimum modification parameters:(3)$$Y_{O}=\beta_{0}+\sum_{j=1}^{3}\beta_{j}X_{j}\sum_{j=1}^{3}\beta_{\mathrm{jj}}X_{j}^{2}+\sum_{j=1}^{3}\sum_{i=j+1}^{3}\beta_{\mathrm{ij}}X_{i}X_{j}$$
where X_i_ and X_j_ are the independent variables, Y is the degree of acetyl substitution, β_ij_ is the interaction term, β_ii_ is the quadratic coefficient, β_i_ is the linear coefficient, and β_0_ is the intercept.

### 2.5. Structural Characterization

#### 2.5.1. Fourier Transform Infrared (FT-IR) Analysis

The dried sample (2 mg) was mixed with KBr (100 mg) powder, and pressed into a pellet. The infrared spectrum was recorded using an FT-IR spectrometer (Gangdong Sci. & Tech. development Co. Ltd., Tianjin, China) in the frequency range of 4000–400 cm^−1^. The number of scans was 32 times.

#### 2.5.2. Monosaccharide Composition Analysis

The monosaccharide composition of the acetylated polysaccharides was determined by 1-pheny-3-methyl-5-pyrazolone (PMP) pre-column derivatization according to a reported reference with some modifications [22]. The sample was mixed with trifluoroacetic acid (TFA, 2 mol/L) and hydrolyzed at 120 °C for 3 h in a hydrothermal synthesis reactor, and then evaporated with anhydrous ethanol to remove excess acid. The hydrolysis product was labeled by successively adding NaOH (0.3 mol/L), PMP (0.5 mol/L), and HCl (0.3 mol/L) solution, and the reaction was carried out in a water bath at 70 °C for 60 min. Finally, the PMP-labeled derivative was analyzed by an HPLC system (Hitachi High-Tech Science Co. Ltd., Tokyo, Japan). The chromatographic conditions: Supersil ODS2 column (5 μm, 4.6 mm × 250 mm), acetonitrile, and phosphate buffer solution (0.02 mol/L, pH 6.8) were used as mobile phase (83:17, *v*/*v*), the column temperature was 35°C, the flow rate was 0.8 mL/min, and the injection volume was 20 μL. The monosaccharide composition and molar ratio of the samples were determined by the retention time of monosaccharide standards and their standard curve. The mannose (Man), glucosamine (GlcN), ribose (Rib), rhamnose (Rha), glucuronicacid (GlcA), galacturonicacid (GalA), glucose (Glc), galactose (Gal), xylose (Xyl), arabinose (Ara), and fucose (Fuc) were used as the monosaccharide standards.

#### 2.5.3. Molecular Weight Analysis

The molecular weight of the sample was measured by high-performance gel permeation chromatography that was equipped with a refractive index detector (HPGPC-RID) [23]. The sample solution was filtered through a 0.45 μm filter membrane, and injected into a Shodex sugar KS-804 column (8.0 mm × 300 mm). The mobile phase was ultrapure water, the column temperature was 50 °C, the flow rate was 1 mL/min, and the injection volume was 20 μL. The chromatogram data were recorded and analyzed using the N2000 GPC system (Hangzhou Sno Scientific Instrument Co. Ltd., Hangzhou, China). The calibration curve was plotted by a series of dextran with different molecular weights.

#### 2.5.4. Congo Red Test

The triple helix conformation of the sample was determined according to previous report [24]. The equal volume of each sample solution (2.5 mg/mL) and Congo red solution (80 μmol/L) were mixed, and NaOH was gradually added to the alkali solution concentrations of 0.1, 0.2, 0.3, 0.4, and 0.5 mol/L, respectively. The mixture alkaline solution without polysaccharides were prepared as the blank control. After equilibrating for 10 min, each maximum absorption wavelength (λ_max_) was measured by UV-vis spectrophotometer (Shanghai Lengguang Technology Co. Ltd., Shanghai, China) in the range of 200~800 nm at room temperature.

### 2.6. Anticomplementary Activity of Polysacchairdes

The anticomplement activity of unmodified and acetylated polysaccharides via classical pathway (CP) and alternative pathway (AP) were determined by referring to previous research methods [25]. Human blood was extracted from normal healthy adults and placed in a rapid coagulation tube. After being placed at 4 °C for 1 h, it was centrifuged at 4 °C for 10 min. The supernatant was taken to obtain normal healthy adult serum pool (NHSP) which was used as the complement source of the classical pathway and stored in a −80 °C refrigerator for standby. Samples with different concentration gradients were diluted with GVB-Ca^2+^/Mg^2+^ as solvent, mixed with NHSP (1:80), and incubated at 37 °C for 30 min. Then, sensitized sheep erythrocytes (EAs, 2.0 × 10^9^ cells/mL) were added and incubated again for 30 min at 37 °C. The reaction mixture was centrifuged to take the supernatant, and the optical density of the supernatant was measured at 540 nm. The hemolysis inhibition rate of polysaccharide samples with different concentrations was calculated, followed by the 50% hemolysis inhibition rate of the classical pathway (CH_50_). Similarly, GVB-Ca^2+^/Mg^2+^ buffer was mixed with fresh rabbit blood, centrifuged in a low-temperature high-speed centrifuge for 5 min, the supernatant was discarded, and 2% rabbit erythrocytes (ERs) were prepared with Aldrin solution as solvent. The 2% Ers were diluted with GVB-Mg^2+^/EGTA to a concentration of 5.0 × 10^8^ cells/mL. NHSP (1:8) in GVB-Mg^2+^/EGTA and different dilutions of polysaccharides were mixed with 2% ERs. The reaction was carried out under 37 °C water bath for 30 min. Finally, the reaction was terminated by cooling on ice. The supernatant was obtained by centrifugation, and the optical density was measured at 412 nm. The 50% hemolytic inhibition through alternative pathways (AP_50_) were used to evaluate the anticomplement activity. Heparin (Activity > 150 USP units/mg) was applied to positive control, which was derived from porcine intestinal mucosa.

### 2.7. Determination of Complement Targets

The minimum concentration of polysaccharides that were required to inhibit hemolysis close to 100% under the CP was selected as the critical concentration of anticomplementary target. The complement-depleted serum (C2, C3, C4, C5, C9) was mixed with sensitized sheep erythrocytes and incubated in 37 °C water bath for 30 min. The absorbance value of the supernatant was measured at 540 nm, and the hemolysis rate was calculated. The minimum concentration of polysaccharides that were required to inhibit hemolysis close to 100% under the AP was selected as the critical concentration of the anticomplement target, and the depleted serum (Factor B, Factor D, and Factor P) and rabbit erythrocytes were added successively. Then, the reaction was incubated at 37 °C for 30 min, and terminated in an ice bath for 5 min. After centrifugation, 30 µL supernatant was taken and added to 270 µL GVB-Mg^2+^/EGTA buffer for dilution. Finally, the absorbance value was measured at 412 nm, and the hemolysis rate was calculated.

### 2.8. Statistical Analysis

All the values were presented as the mean ± standard deviation (*n* = 3). SPSS 17.0 statistical software (SPSS Inc., Chicago, JOT, USA) carried out to calculate the CH_50_ and AP_50_ of biological activity. Statistical significances were calculated by one-way analysis of variance (ANOVA) using GraphPad Prism 6.0 (GraphPad Software Inc., San Diego, CA, USA), and the value of *p* < 0.05 was considered to be significant.

## 3. Results and Discussion

### 3.1. Single Factor Analysis

The degree of substitution (DS) was calculated according to the acetyl standard curve equation (y = 7.1499x − 0.0207, *R*^2^ = 0.9993). As shown in Figure 1a, with the increase of reaction time, the DS of polysaccharides increased rapidly in the range of 1 to 3 h (*p* < 0.05), which improved the interaction rate between the polysaccharides and the acetylation reagent, and was more conducive to the occurrence of nucleophilic substitution reaction. When the reaction time was more than 3 h, the DS of polysaccharides decreased slightly and stabilized, which might because the acetyl group that had been successfully substituted was unstable in the alkaline environment and was cracked with the prolonged reaction time. Figure 1b shows the effect of the reaction temperature on acetylation modification of polysaccharides; the DS of polysaccharides acetylation modification gradually increased with the increase of reaction temperature in the range of 30 °C to 40 °C (*p* < 0.05). It is worth noting that the DS of polysaccharides had no significant change in the range of 40 °C to 60 °C (*p* > 0.05), indicating that the acetyl substitution reaction of polysaccharides was relatively stable in this range. However, the DS of polysaccharides tended to decrease when the temperature was higher than 60 °C (*p* < 0.05), which was attributed to the fact that the hydroxyl groups of polysaccharides were not easily exposed and the substitution reaction was restricted in the high temperature environment. As shown in Figure 1c, the effect of the ratio of acetic anhydride solution addition amount to the weight of polysaccharides sample (i.e., liquid-solid ratio) on the modification substitution degree. When the liquid-solid ratio was in the range of 8 mL/g to 16 mL/g, the acetylation modification of polysaccharides increased with the increase of the liquid-solid ratio, which can also be interpreted as the increase of acetic anhydride addition in the modification reaction system increased the chance of interaction between the acetyl reagent and polysaccharides (*p* < 0.05). When the liquid-solid ratio exceeded 16 mL/g, the acetylation substitution of the polysaccharides decreased instead. The reason for the decrease in substitution may be that the excess of acetic anhydride increased the extent of hydrolysis side reactions in the synthetic system and the already acetylated polysaccharides inhibited the continuation of the acetylation reaction.

### 3.2. RSM Optimization of Acetylation Modification

#### 3.2.1. Model Fitting

Based on single factor experimental results, the reaction time of 1 to 5 h, reaction temperature of 40 to 60 °C and liquid-solid ratio of 12 to 20 mL/g were employed to further optimize the modification conditions of RDPs via BBD with 17 runs, and the actual and predicted values of DS are displayed in Table 1.

According to the multiple regression analysis of experimental data, the response value and independent variables are expressed as quadratic polynomials:Y = 0.47 + 0.011 × X_1_ −0.00009199 × X_2_ − 0.011 × X_3_ − 0.013 × X_1_ × X_2_ + 0.11 × X_1_ × X_3_ − 0.087 × X_2_ × X_3_ − 0.069 × X_1_^2^ − 0.068 × X_2_^2^ − 0.15 × X_3_^2^(4)

As shown in Table 2, the statistical differences of the model were evaluated by ANOVA. The model F-value was 238.69 and *p*-value was <0.0001, which implied that the model was significant. The lack of fit F-value (2.25) and *p*-value (0.2242) demonstrated the lack of fit is not significant relative to the pure error. The coefficient of determination (*R^2^*), the adjusted coefficient determination (Adj-*R^2^*), the predicted coefficient determination (Pred *R^2^*), and the coefficient of variation (C.V. %) in the regression model were 0.9968, 0.9926, and 3.06, respectively, which suggested that the model had excellent correlation between the predicted value and the experimental value [26]. The *p*-values of linear coefficients X_1_ and X_3_ (*p* < 0.05), quadratic coefficients X_1×2_ were significantly (*p* < 0.05), and the *p*-values of other coefficients (X_1_X_3_, X_2_X_3_, X_1_^2^, X_2_^2^, and X_3_^2^) were extremely significant (*p* < 0.01), which indicated that the reaction time and liquid-solid ratio had a strong correlation with the substitution degree of acetylated RDPs. However, the *p*-value of X_2_ indicated that the correlation between the reaction temperature and polysaccharide modification was poor within the range of 40 to 60 °C (*p* > 0.05), which was consistent with the above single factor experiment results.

#### 3.2.2. Response Surface Analysis

Three-dimensional (3D) and two-dimensional contour plots (2D) of the response surface provide an initial visualization of the effect of each variable on the degree of polysaccharides acetyl substitution and the interaction between the two variables. As shown, Figure 2b,c showed steeper response surfaces than other 3D picture, meanwhile Figure 2d,e exhibited more elliptical contour lines than other 2D plot, which indicated that interaction effects of reaction time and liquid-solid ratio, and the reaction temperature and liquid-solid ratio had extremely significant differences on DS of RDPs. In addition, it can be seen that there was a peak of each response surface graph, which also indicated that the setting of variable range was reasonable.

#### 3.2.3. Verification of the Predictive Model

According to multiple regression analysis and model prediction, the optimal acetylation modification conditions of RDPs were obtained by Design-Expert 8.0.6 statistical software as follows: a reaction time of 3.15 h, reaction temperature of 49.98 °C, and a liquid-solid ratio of 15.96 mL/g. Under these conditions, the predicted DS was 0.472. Considering the actual operation process, the optimal parameters were as follows: reaction time of 3 h, reaction temperature of 50 °C, liquid-solid ratio of 16 mL/g, the actual DS of RDPs was 0.465 ± 0.014 (*n* = 3), which was close to the predicted value, indicating the accuracy and reliability of the optimization model for the acetylation modification process of RDPs. Furthermore, two acetylated polysaccharide fractions (AcRDP-1 and AcRDP-2) were obtained by acetylation modification of two fractions from *R. dauricum* leaves (RDP-1 and RDP-2) under this optimal condition. The results showed that the DS and modification yields of AcRDP-1 and AcRDP-2 were 0.439 ± 0.025 and 73.08 ± 1.82%, 0.445 ± 0.022, and 78.32 ± 2.13%, respectively. In addition, the extraction yields of RDP-1 and RDP-2 were 13.21 ± 1.22% and 9.87 ± 1.04%, respectively, which were similar to polysaccharides from leaves that were reported in previous studies [27,28,29], and provided a relatively stable source of raw materials for the industrial production of polysaccharides.

### 3.3. Structural Characterization of AcRDP-1 and AcRDP-2

#### 3.3.1. FT-IR Analysis

The FT-IR spectra of two acetylated polysaccharides (AcRDP-1 and AcRDP-2) were presented in Figure 3. The strong absorptions at 3400 cm^−1^ and 2930 cm^−1^ were attributed to the stretching vibration of O-H and C-H, respectively. The peak at around 1614 cm^−1^ was assigned to bound water [30]. The peak at approximately 1370 cm^−1^ was related to the bending vibration of C-H [31]. The signal peak at about 1070 cm^−1^ was ascribed to the symmetric stretching vibration of the C-O-C [32]. Compared to the unmodified polysaccharides that were reported in our previous study [3], the new absorption peak that was found at about 1740 cm^−1^ was attributed to the vibration of C=O of the O-acetyl group, and the enhanced absorption peak at around 1245 cm^−1^ was caused by the asymmetric stretching vibration of C-O-C in carbonyl groups, which proved that the two acetylated polysaccharides were successfully modified [6,33].

#### 3.3.2. Monosaccharide Composition Analysis

As shown in Figure 4, AcRDP-1 and AcRDP-2 were composed of Man, Glc, Gal, and Ara with molar ratios of 1.00:5.01:1.17:0.15 and 1.00:4.47:2.39:0.88, respectively. The results showed that the monosaccharide types of the two acetylated polysaccharides were the same as those of the unmodified polysaccharides in the previous study [3], and they consisted of four monosaccharides. However, the molar ratio of each monosaccharide of acetylated polysaccharides was changed. Among that, the content of glucose remained the highest, while the content of arabinose decreased, and the molar ratio of mannose to galactose was different from that of the unmodified polysaccharides. It was speculated that it might be due to the breakage of glycosidic chains in the preparation of acetylated polysaccharides, and the substitution of several arabinose units by acetyl groups [6].

#### 3.3.3. Molecular Weight Analysis

The HPGPC of AcRDP-1 and AcRDP-2 exhibited a single and symmetrical peak, indicating that AcRDP-1 and AcRDP-2 were homogenous polysaccharides (Figure 5). The molecular weight (Mw) of AcRDP-1 and AcRDP-2 were calculated to be 9.3525 × 10^3^ kDa and 4.7016 × 10^3^ kDa, respectively. Compared with the unmodified polysaccharides (RDP-1: 1.0948 × 10^4^ kDa and RDP-2: 1.1899 × 10^5^ kDa), the Mw of the two acetylated polysaccharides decreased significantly, which was due to the degradation of the polysaccharides under the action of acetylating agents. Similarly, several studies also found that the Mw of acetylated derivatives is significantly reduced during the modification process [19,34,35]. Interestingly, the decrease rate of molecular weight of AcRDP-2 (96.04%) was higher than that of ACRDP-1 (14.57%), suggesting that AcRDP-2 may be more susceptible to degradation.

#### 3.3.4. Congo Red Test

The Congo red test is a common method to evaluate the existence of triple helix conformation in polysaccharides. The complex that is formed by polysaccharide with triple helical conformation and Congo red solution, and the maximum absorption wavelength (λ_max_) of the complex can be red shifted with the increase of concentration of NaOH solution [36]. However, as shown in Figure 6, the change trend of AcRDP-1 and AcRDP-2 was the same as that of the blank group. With the increase of NaOH concentration in the solution, the λmax continued to decline, and no complex was formed with Congo red, indicating that AcRDP-1 and AcRDP-2 had no triple helix conformation. It can be seen that the triple helix conformation of polysaccharides disappeared through acetylation modification, which may be caused by the breakage of glycosidic chains and the change of the force between glycosidic bonds in the modification process.

### 3.4. Anticomplementary Activity Analysis

The anticomplementary activity of unmodified polysaccharides (RDP-1 and RDP-2) and acetylated polysaccharides (AcRDP-1 and AcRDP-2) were evaluated on the CP and AP of the complement system, and their 50% hemolytic inhibition concentrations (CH_50_ and AP_50_) are shown in Table 3. Compared with RDP-1, AcRDP-1 produced anti-complement activity through acetylation modification, and the CH_50_ and AP_50_ values of AcRDP-1 were significantly lower than those of the positive control (*p* < 0.05), indicating that ACRDP-1 had a strong complement inhibitory effect. Although RDP-2 showed weak complement inhibition effect in the CP, its CH_50_ value was significantly higher than that of heparin (*p* < 0.05), and it was inactive in the AP. Excitingly, the CH_50_ and AP_50_ of AcRDP-2 were also significantly lower than the positive drugs (*p* < 0.05), suggesting that AcRDP-2 produced a stronger new anticomplementary activity than the positive drug through acetylation modification. Compared with other polysaccharides that were previously reported [16,17,18], AcRDP-1 and AcRDP-2 have stronger complement inhibition ability in the CP and AP, which indicated that the acetylated polysaccharides from *R. dauricum* leaves have higher development potential.

Ulteriorly, the complement-depleted serum was used to identify AcRDP-1 and AcRDP-2 targets in the complement activation cascade. As shown in Figure 7a, AcRDP-1 did not restore the hemolytic activity in C2-, C3-, C4-, C5-, C9-, Factor B-, or Factor D-depleted sera (the hemolysis percentage was less than 20%), while it did obviously restore hemolysis in Factor P-depleted sera. Thus, AcRDP-1 interacted with C2, C3, C4, C5, C9, Factor B, and Factor D targets. Meanwhile, as shown in Figure 7b, AcRDP-2 interacted with complement components C2, C3, C4, C5, C9, and Factor B to inhibit the overactivation of complement, but did not block the targets of Factor D and Factor P. Combined with the above results, AcRDP-1 and AcRDP-2 showed excellent complement inhibition ability via the CP and AP with multiple targets, which demonstrated the crucial role of acetylation modification in blocking complement components.

On the whole, the anticomplementary activity of AcRDP-1 and AcRDP-2 was effectively increased by acetylation modification, which was associated with changes in the structural characteristics of AcRDP-1 and AcRDP-2. The activity of large molecular weight polysaccharides was susceptible to spatial hindrance [37], while the lower molecular weight acetylated polysaccharides may be more conducive to blocking complement components. Also, a large number of -OH groups were substituted by CH3-CO- groups in the polysaccharides with high DS. Huo et al. [18] stated that the presence of uronic acid in polysaccharides was considered important. However, unmodified polysaccharides (RDP-1 and RDP-2) do not have uronic acid, which may be one of the factors that cause the anticomplementary activity of RDP-1 and RDP-2 to be significantly weaker than that of heparin. With the introduction of acetyl groups, the triple helix conformation was de-rotated, and the extensibility of the molecular chain was improved, resulting in the enhancement of complement inhibition effect [4].

Heparin is a highly sulfated glycosaminoglycan that is widely used as an anticoagulant, which is alternately composed of glucosamine, L-eduraldehyde glycoside, N-acetylglucosamine, and D-glucuronic acid with linear chains, and the average molecular weight is about 15 kDa. In the early years, many scholars found that heparin could act on the classical and bypass pathways of complement and has long been considered as an inhibitor of complement activation in vitro [38,39]. It interferes with complement activation at multiple levels, by binding and inactivating C1, blocking the assembly of the C3 convertases, and interfering with the assembly of membrane attack complex (MAC) [40,41,42,43]. However, the application of heparin to complement inhibition in vivo is flawed due to its anticoagulant properties [44]. Compared with heparin, the RDPs and AcRDPs have completely different monosaccharide compositions, molecular weights, and chain extensibility, which is the reason for their different anticomplementary ability and mechanism. Heparin induced in vivo anticoagulation mainly through the ionic interaction of its sulfate group with amino acid residues on coagulation factors [45,46], whereas RDPs and AcRDPs did not have sulfate groups, which implied that RDPs and AcRDPs may have less side effects than heparin in vivo. Thus, RDPs and AcRDPs have significant advantages in the treatment of complement hyperactivation-related diseases.

## 4. Conclusions

In this study, the optimal acetylation modification parameters of *R. dauricum* leaves polysaccharides (RDPs) were as follows: a reaction time of 3 h, reaction temperature of 50 °C, and a liquid-solid ratio of 16 mL/g. Under these conditions, two acetylated polysaccharides from *R. dauricum* leaves (AcRDP-1 with DS of 0.439 ± 0.025 and AcRDP-2 with DS of 0.445 ± 0.022) were successfully obtained. The results of structural characterization showed that both AcRDP-1 and AcRDP-2 were composed of mannose, glucose, galactose, and arabinose with obvious acetyl characteristic groups and lower molecular weight, but no triple helix configuration. The two acetylated polysaccharides exhibited a stronger complement inhibitory effect than positive drugs through the classical pathway and alternative pathway by blocking C2, C3, C4, C5, C9, and Factor B targets, which demonstrated their potential development as novel anticomplementary drugs in the pharmaceutical industry. Therefore, this study provided an optimal process for acetylation modification of RDPs and successfully improved the biological activities of RDPs, which provided a new direction for the utilization of *R. dauricum* leaves and laid a theoretical foundation for its industrial production. However, acetyl group substitution sites and more explicit structure-activity relationships of acetylated polysaccharides need to be further studied in the future.

## Figures and Tables

**Figure 1 polymers-14-03130-f001:**
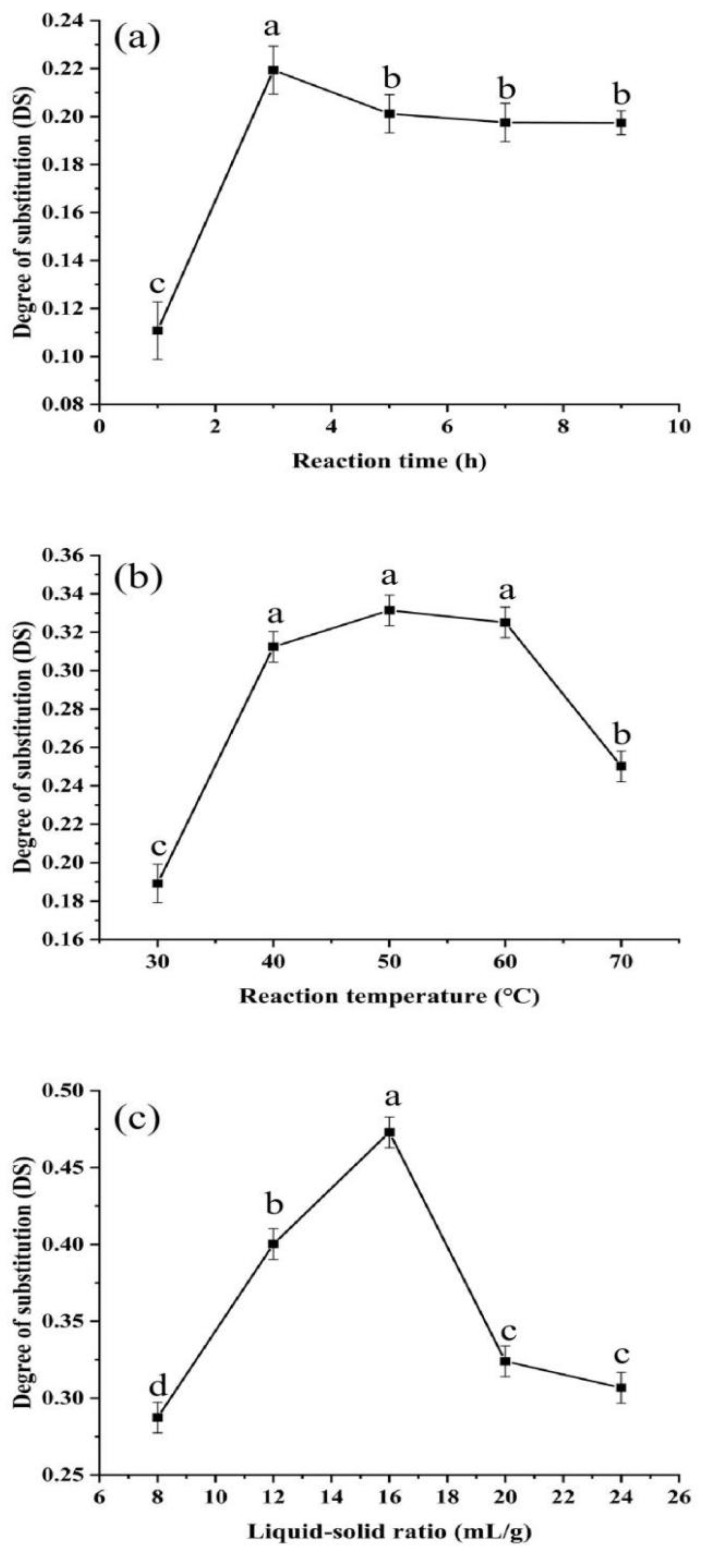
Effect of reaction time (**a**), reaction temperature (**b**), and liquid-solid ratio (**c**) on the degree of substitution of RDPs. Different letters indicate significant differences (*p* < 0.05).

**Figure 2 polymers-14-03130-f002:**
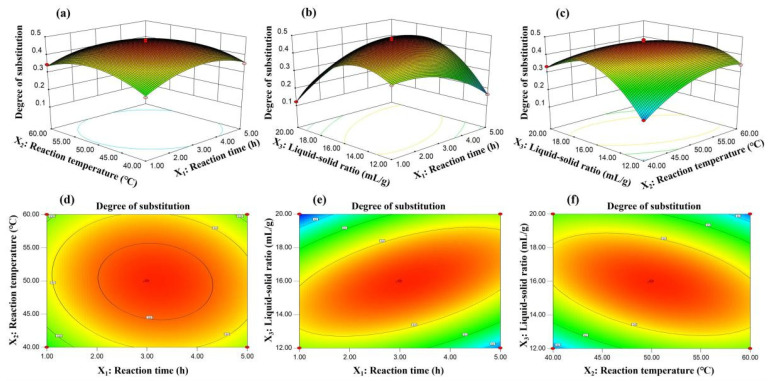
Three-dimensional (3D) and two-dimensional contour plots (2D) of the response surface and their mutual interactions on the degree of substitution of RDPs. (**a**,**d**) The reaction time/reaction temperature; (**b**,**e**) the reaction time/liquid-solid ratio; (**c**,**f**) the reaction temperature/liquid-solid ratio).

**Figure 3 polymers-14-03130-f003:**
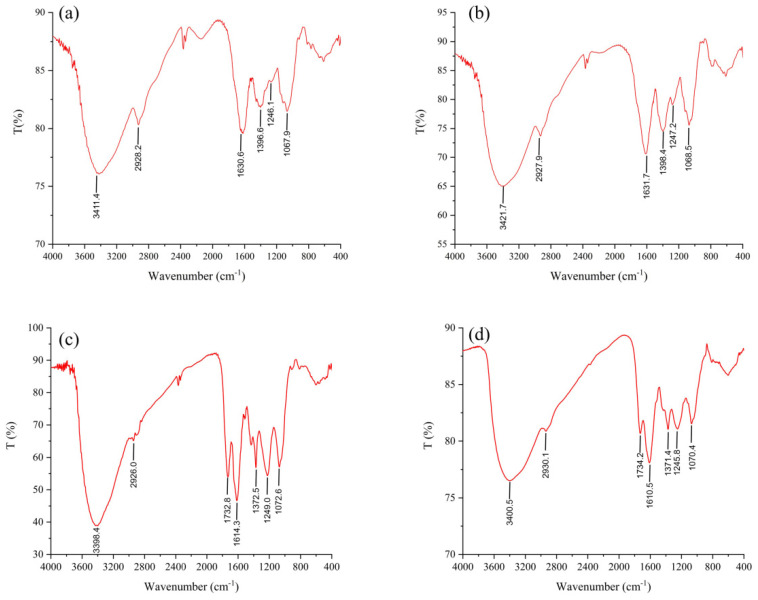
Fourier transform infrared spectrogram of RDP-1 (**a**), RDP-2 (**b**), AcRDP-1 (**c**), and AcRDP-2 (**d**).

**Figure 4 polymers-14-03130-f004:**
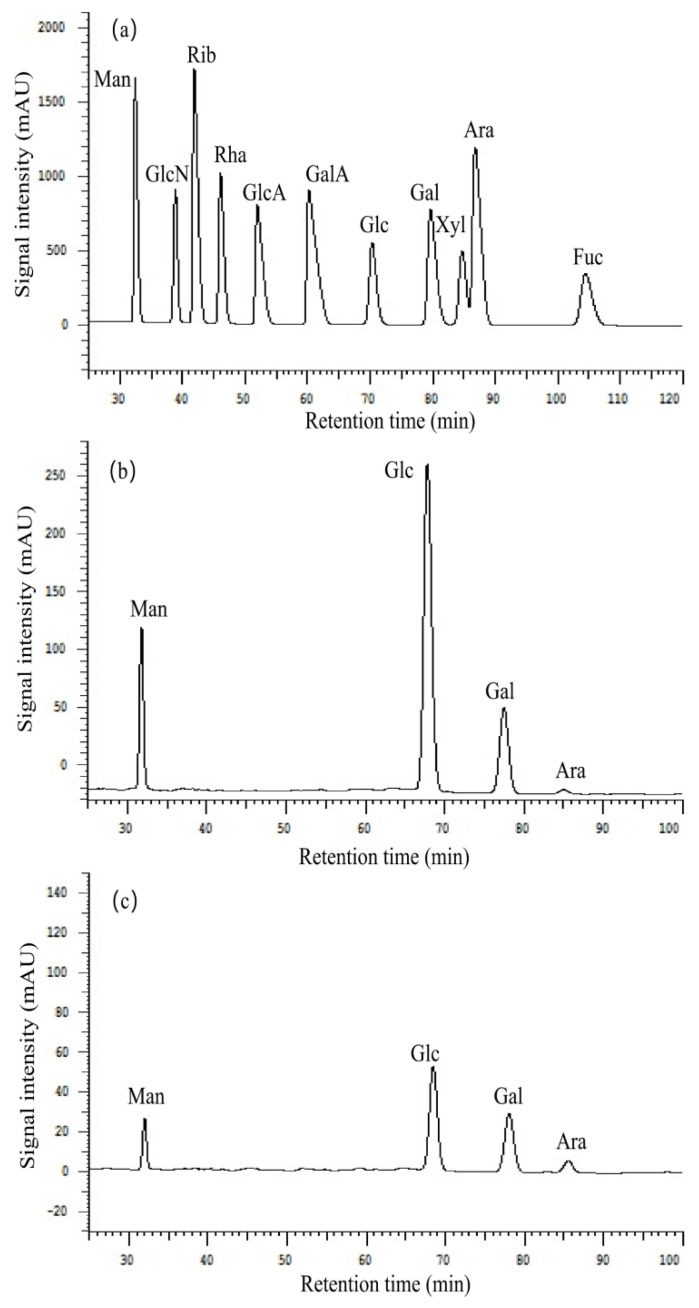
High performance liquid chromatogram of monosaccharide composition of standard monosaccharide (**a**), AcRDP-1 (**b**), and AcRDP-2 (**c**). (Man, mannose; GlcN, glucosamine; Rib, ribose; Rha, rhamnose; GlcA, glucuronicacid; GalA, galacturonicacid; Glc, glucose; Gal, galactose; Xyl, xylose; Ara, arabinose; Fuc, fucose).

**Figure 5 polymers-14-03130-f005:**
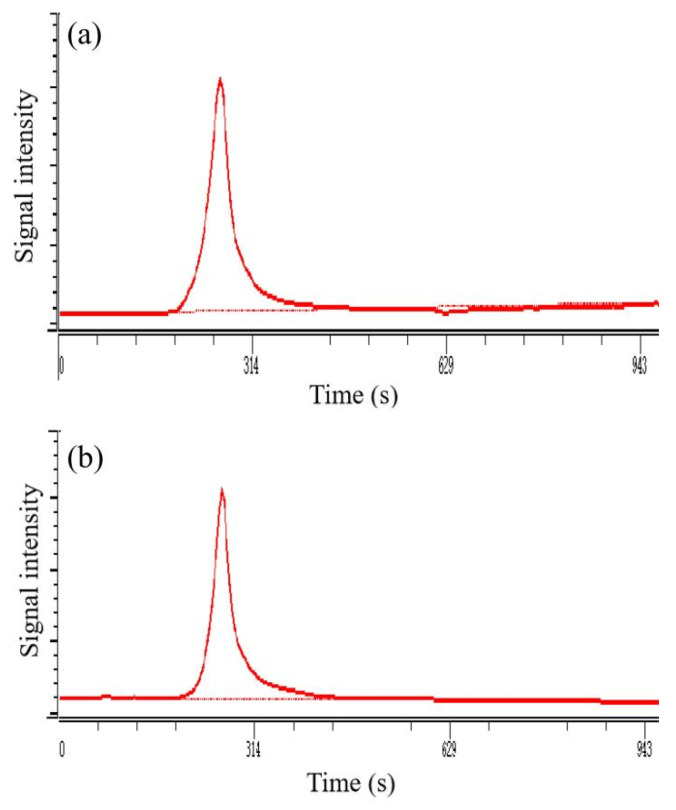
High performance gel permeation chromatography of AcRDP-1 (**a**) and AcRDP-2 (**b**).

**Figure 6 polymers-14-03130-f006:**
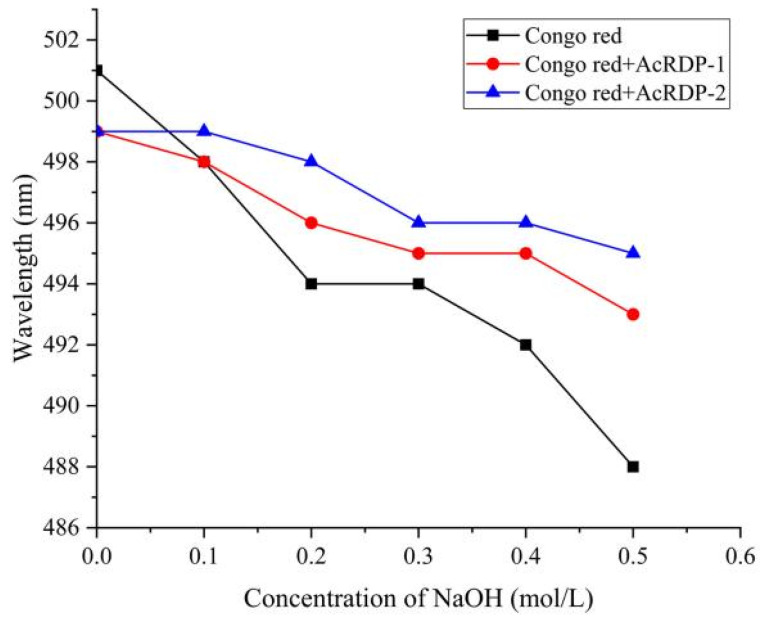
Changes in the absorption wavelength maximum of Congo red with AcRDP-1 and AcRDP-2 complexes at various concentrations of NaOH.

**Figure 7 polymers-14-03130-f007:**
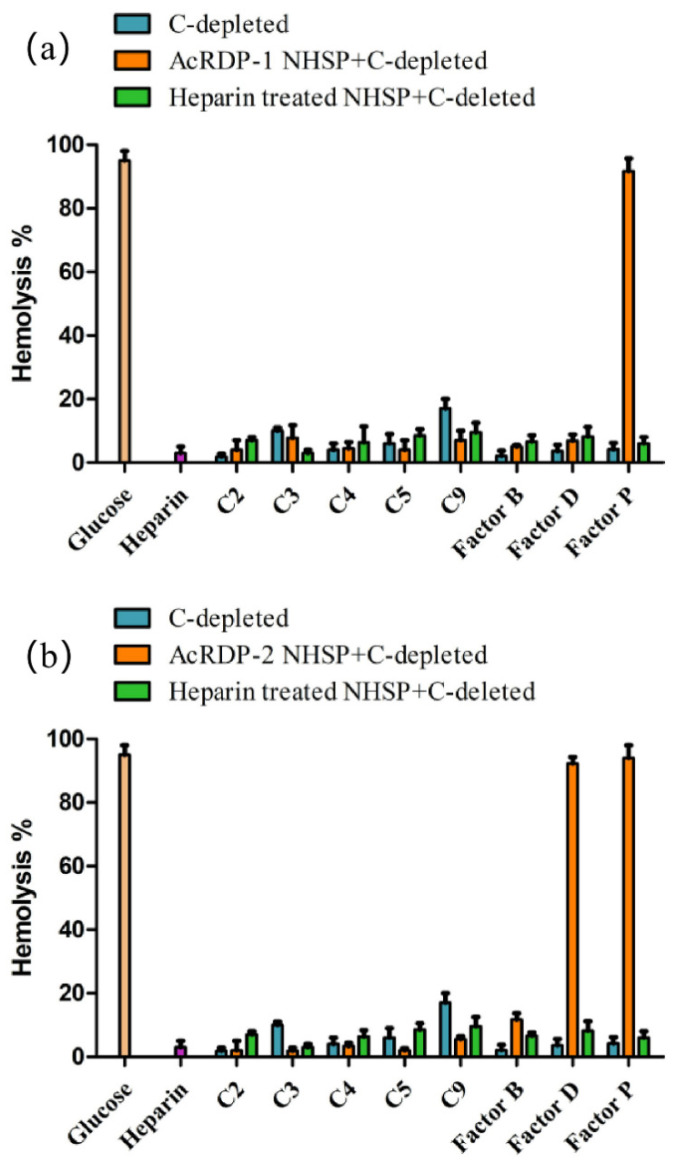
Targets of AcRDP-1 (**a**) and AcRDP-2 (**b**) in the complement activation cascade.

**Table 1 polymers-14-03130-t001:** Experimental design and the results of the Box–Behnken design.

Run	Reaction Time (X_1_, h)	Reaction Temperature (X_2_, °C)	Liquid-Solid Ratio (X_3_, mL/g)	Degree of Substitution	
				Actual Value	Predicted Value
1	3 (0)	60 (1)	12 (−1)	0.34	0.34
2	5 (1)	50 (0)	12 (−1)	0.16	0.17
3	1 (−1)	60 (1)	16 (0)	0.34	0.35
4	3 (0)	50 (0)	16 (0)	0.46	0.47
5	3 (0)	50 (0)	16 (0)	0.47	0.47
6	5 (1)	40 (−1)	16 (0)	0.35	0.35
7	5 (1)	50 (0)	20 (1)	0.36	0.36
8	5 (1)	60 (1)	16 (0)	0.34	0.33
9	1 (−1)	40 (−1)	16 (0)	0.30	0.31
10	3 (0)	60 (1)	20 (1)	0.14	0.15
11	3 (0)	50 (0)	16 (0)	0.46	0.47
12	3 (0)	50 (0)	16 (0)	0.481	0.47
13	1 (−1)	50 (0)	12 (−1)	0.35	0.36
14	3 (0)	40 (−1)	20 (1)	0.33	0.33
15	3 (0)	50 (0)	16 (0)	0.47	0.47
16	1 (−1)	50 (0)	20 (1)	0.12	0.12
17	3 (0)	40 (−1)	12 (−1)	0.19	0.18

**Table 2 polymers-14-03130-t002:** Analysis of variance for response surface quadratic model.

Source	Sum of Squares	Df	Mean Square	F-Value	*p*-Value
Model	0.23	9	0.025	238.69	<0.0001 **
X_1_	0.001033	1	0.001033	9.74	0.0168 *
X_2_	0.0000000677	1	0000000677	0.0006385	0.9805
X_3_	0.0009834	1	0.0009834	9.27	0.0243 *
X_1_X_2_	0.0007224	1	0.0007224	6.81	0.0349 *
X_1_X_3_	0.047	1	0.047	438.82	<0.0001 **
X_2_X_3_	0.030	1	0.030	283.14	<0.0001 **
X_1_^2^	0.020	1	0.020	187.10	<0.0001 **
X_2_^2^	0.019	1	0.019	181.79	<0.0001 **
X_3_^2^	0.097	1	0.097	911.72	<0.0001 **
Residual	0.0007422	7	0.022	—	—
Lack of Fit	0.0004664	3	0001555	2.25	0.2242
Pure Error	0.0002758	4	0.00006896	—	—
Cor Total	0.23	16	—	—	—
*R^2^*	0.9968	—	Adj *R^2^*	0.9926	—
C.V. %	3.06	—	Pred *R^2^*	0.9655	—

* Indicates significant differences (*p* < 0.05); ** Indicates extremely significant differences (*p* < 0.01).

**Table 3 polymers-14-03130-t003:** Anticomplementary activities of unmodified and acetylated polysaccharides.

Samples	CH_50_ (mg/mL)	AP_50_ (mg/mL)
RDP-1	NE	NE
RDP-2	0.870 ± 0.030 ^a^	NE
AcRDP-1	0.009 ± 0.003 ^c^	0.015 ± 0.003 ^b^
AcRDP-2	0.004 ± 0.001 ^c^	0.028 ± 0.005 ^b^
Heparin	0.114 ± 0.013 ^b^	0.138 ± 0.012 ^a^

Data are expressed as the mean ± SD (n = 3); CH_50_ and AP_50_ stand for the 50% hemolytic inhibition concentrations through the classical pathway and alternative pathway, respectively. Heparin was positive control for anticomplementary activity. “NE” denotes that this sample has no inhibitory effect at the maximal concentration that was tested. Different letters (a–c) indicate significant differences in the same complement pathway (*p* < 0.05).

## Data Availability

The data presented in this study are available on request from the corresponding author.

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
