# Peer review of "Acetylation Modification, Characterization, and Anticomplementary Activity of Polysaccharides from Rhododendron dauricum Leaves"

_polymers, 2022, doi:10.3390/polym14153130_

Round 1

Reviewer 1 Report

The manuscript describes a study of the anticomplementary activity of polysaccharides extracted from Rhododendron leaves after purification and acetylation.

1    The purpose of the work was investigation of relatively pure polysaccharides in order to assist interpretation of structural and biological activity studies. The samples used were of known molecular weight and degree of acetyl substitution. The critical comparison with respect to anticomplement activity in Table 3, however, was with a control mentioned as Heparin in the last sentence of  section 2.6.

A more complete description of the Heparin control is needed since it can have many sources. A discussion should also be included regarding the mechanism of anticomplement activity for Heparin as compared to what is expected for RDP-1 and -2 related to factors such as molecular weight, chain extensibility and composition.

2    It should also be noted that the CH50 and AP50 values are quite low. A typical CH50 for acidic polysaccharide seems to be about 0.3-0.5 mg/ml. Since human serum complement is a sensitive product subject to instability, its source and processing history should be given in section 2.6. A few references should also be added describing anticomplement activity for other polysaccharides in order to give perspective to the values shown in Table 3.

3    Related to the above is the report of a low CH50 of 0.08 mg/ml in J Huo et al., Structural characterization and anticomplement activity of an acidic polysaccharide from Hedyotis diffusa, International Journal of Biological Macromolecules 155 (2020) 1553-1560. The presence of uronic acid was stated to be important. This conclusion should  be discussed since reference 3 makes it clear that RDP-1 and -2 do not contain the acid, whereas heparin does.

4    Purification separates the polysaccharides from their traditional medicine history where several components are included. No information is available about the anticomplement activity of the crude extract and R. dauricum is thus only a source of relatively inactive polysaccharide starting material in the study.

There are two aspects that should be discussed. One is if RDP-1 and -2, irrespective of acetylation, have any specific medicinal advantages based on their mannose glucose galactose arabinose composition.  

The other concerns the amount of polysaccharides which can be produced from the leaves. The RDP-1 and -2 extraction yields should be reported and some discussion included about if the percentages justify the statement about industrial production on line 423.

5    It seems unnecessary to change the indexing of the (unacetylated) composition from mannose at 1 in reference 3 to arabose at 1 after acetylation in the manuscript.   

Reviewer 2 Report

A manuscript by Hu and colleagues presents the acetylation modification of polysaccharides from rhododendron leaves, which could be useful for various biomedical applications. The manuscript is well developed, and their results represent an important contribution to the improvement of materials science in general. I have a few minor suggestions that the authors might consider, but none of them would prevent moving forward:

1)      Figure 3: I recommend that the authors include the FT-IR spectrogram of the unmodified polysaccharide for clarity, as they are compared to the unmodified polysaccharide in the text.

2)      I suggest that the authors also investigate the antioxidant effects of both AcRDPs on cell lines (e.g., pulmonary epithelial cells), which would add more scientific value to the manuscript.

Round 2

Reviewer 1 Report

All issues resolved.